# Fractals as Julia Sets of Complex Sine Function via Fixed Point Iterations

**Swati Antal** [1] , **Anita Tomar** [2] , **Darshana J. Prajapati** [3] **and Mohammad Sajid** [4,*]

1   BGR Campus, Hemavati Nandan Bahuguna Garhwal University, Pauri Garhwal 246001, India; antalswati11@gmail.com
2   Pt. L.M.S. Campus, Sri Dev Suman Uttrakhand University, Rishikesh 249201, India; anitatmr@yahoo.com
3   M.B. Patel Institute of Technology, New Vallabh Vidyanagar 388121, India; djprajapati@mbit.edu.in
4   Department of Mechanical Engineering, College of Engineering, Qassim University, Buraydah 51452, Saudi Arabia
*   Correspondence: msajd@qu.edu.sa

**Abstract:** We explore some new variants of the Julia set by developing the escape criteria for a function $\sin(z^n) + az + c$, where $a, c \in \mathbb{C}$, $n \geq 2$, and $z$ is a complex variable, utilizing four distinct fixed point iterative methods. Furthermore, we examine the impact of parameters on the deviation of dynamics, color, and appearance of fractals. Some of these fractals represent the stunning art on glass, and Rangoli (made in different parts of India, especially during the festive season) which are useful in interior decoration. Some fractals are similar to beautiful objects found in our surroundings like flowers (to be specific Hibiscus and Catharanthus Roseus), and ants.

**Keywords:** escape criterion; fixed point; Mann orbit; Ishikawa orbit; Noor orbit

## 1. Introduction

Due to the self-similarity, the study of fractals emerged as the most fascinating area of research [1,2]. The investigation on fractals commenced at the beginning of the twentieth century, when French mathematician Gaston Maurice Julia and French mathematician and astronomer Pierre Joseph Louis Fatou studied complex dynamics and determined the successive approximations of the complex function $f(z) = z^2 + c$, $c$ is a complex number and $z$ is a complex variable. In 1918, Julia [3,4] was successful at iterating this function but could not sketch it. Fatou [5] was the first to study the Julia sets. This is why its complement is often known as the Fatou set. It comprises values possessing the characteristic that all neighboring values act in the same way under repeated iteration of the underlying function. On the other hand, the Julia set comprises values having the property that even an arbitrarily small perturbation may create serious changes in the sequence of iterated functions. Consequently, the function on the Fatou set behaves in a "regular" manner, on the other hand, the function on the Julia set behaves in a "chaotic" manner. Later on, around 1980, a Polish-born French-American mathematician and polymath Benoit B. Mandelbrot [1] drew the Julia set and investigated its characteristics. He gave the name fractals to these complex graphs and noticed that, for distinct values of the parameter $c$, the Julia sets possess distinctiveness in their characteristics. In addition, on interchanging the position of $z$ and $c$, he introduced a new set widely known as the Mandelbrot set in which we investigate the connectedness of the Julia set for every value of $c$. On the contrary, in the Julia set, we investigate the behavior of the iterates for every value of $z$. In 1987, Lakhtakia et al. [6] extended their work by utilizing $f(z) = z^n + c$, $n \geq 2$, and, in 1989, Crowe et al. [7] sketched complex graphs for $z^2 + c$ and introduced the anti-Julia and anti-Mandelbrot sets to discuss their connected locus. These graphs are termed as "tricorn" [8]. Since then, numerous generalizations of fractals have come into

the existence via different fixed point iterative methods (for instance, Mann-iteration [9], Ishikawa-iteration [10], $S$-iteration [11], Noor-iteration [12], Jungck-type iterations [13–19], $CR$-iteration [20], $SP$-iteration [21], and so on). Fractals for transcendental complex functions were studied in [22–28]. It is interesting to mention here that an iteration method, also known as a method of successive approximations, permits finding a solution with a predetermined accuracy and may be easily programmed. Consequently, it is convenient to use in computer calculations. In addition, this reduces the time of calculation and is self-correcting because the small errors that are committed in the calculation process of successive approximations may be corrected later. Iteration methods have many real-life applications. For instance, the Hybrid–Euler method [29] which is a blend of Euler's method [30] and the spline interpolation method (suitable for smooth functions which do not have oscillating behavior [31]) has been utilized to predict the total number of infected people and the number of active cases for COVID-19 propagation to describe the dynamics of the pandemic.

In the current work, we explore some novel variants of the Julia set for the transcendental complex function of the type $\sin(z^n) + az + c$, $n \geq 2$, $a$ and $c$ are complex numbers, and $z$ is a complex variable via utilizing four distinct celebrated fixed point iterative methods. First, we develop the escape criteria in algorithms for this transcendental complex sine function, and then we utilize it to sketch Julia sets. It is worth mentioning here that the algorithms perform a significant role in sketching the fractals while, the escape limitations are fundamental requirements for running the algorithms. Finally, we compare the sketched complex graphics in Picard-orbit, Mann-orbit, Ishikawa-orbit, and Noor-orbit and examine the response of stunning Julia sets for distinct variables via these distinct algorithms.

## 2. Preliminaries

The filled Julia set [2,3] of the polynomial $p : \mathbb{C} \to \mathbb{C}$ over the set of complex numbers of degree $\geq 2$ is defined as

$$F_p = \{z \in \mathbb{C} : |p(z_k)|_{k=0}^{\infty} \text{ is bounded}\}.$$

Noticeably, it is a set of complex numbers for which the orbits do not converge to a point at infinity. The Julia set of $p$ is the boundary of $Jp$, that is, $Jp = \partial F_p$.

**Definition 1.** *Let $T : \mathbb{C} \to \mathbb{C}$ be a complex valued self-map. For $z_0 \in \mathbb{C}$, $\alpha, \beta, \gamma \in (0, 1]$ and $k = 0, 1, 2, \ldots$,*

*(i)　the Picard-iteration [32] is*

$$z_{k+1} = T(z_k).$$

*Clearly, it is a one-step feedback procedure.*

*(ii)　the Mann-iteration [9] is*

$$z_{k+1} = (1 - \alpha)z_k + \alpha T(z_k).$$

*Clearly, it is a one-step feedback procedure.*

*(iii)　the Ishikawa-iteration [10] is*

$$\begin{aligned} z_{k+1} &= (1 - \alpha)z_k + \alpha T(y_k), \\ y_k &= (1 - \beta)z_k + \beta T(z_k). \end{aligned}$$

*Clearly, it is a two-step feedback procedure.*

*(iv)　the Noor-iteration [12] is*

$$\begin{aligned} z_{k+1} &= (1 - \alpha)z_k + \alpha T(y_k), \\ y_k &= (1 - \beta)z_k + \beta T(x_k), \\ x_k &= (1 - \gamma)z_k + \gamma T(z_k). \end{aligned}$$

*Clearly, it is a three-step feedback procedure.*

**Remark 1.** *The Noor-orbit diminishes to:*

1. *Ishikawa-orbit when $\gamma = 0$,*
2. *Mann-orbit when $\beta = \gamma = 0$.*
3. *Picard-orbit when $\alpha = 1, \beta = \gamma = 0$.*

Motivated by the applications of the function sine, for instance, in modeling periodic phenomena like the velocity and position of harmonic oscillators, light and sound waves, day length and sunlight intensity, average temperature variations throughout the year, and so on, we utilize sine function to explore some fascinating fractals via four different fixed point iterations. The sine function can be traced to the jyā functions utilized by Indian mathematicians and Astronomers (Aryabhatiya, Surya Siddhanta). However, jyā is a function of arcs of circles instead of angles. The modern trigonometric function of sine is a function of an angle whose only real fixed point is zero. The more improved version expresses the sine as an infinite series, allowing their extension to arbitrary negative, positive, and complex numbers.

It is well known that $|\sin(z^n)| \leq 1, \forall \, z \in \mathbb{C}$ and

$$
\begin{aligned}
|\sin(z^n)| &= \left| z^n - \frac{z^{3n}}{3!} + \frac{z^{5n}}{5!} - \ldots \right| \\
&= |z^n| \left| 1 - \frac{z^{2n}}{3!} + \frac{z^{4n}}{5!} - \ldots \right| \\
|\sin(y^n)| &= \left| y^n - \frac{y^{3n}}{3!} + \frac{y^{5n}}{5!} - \ldots \right| \\
&= |y^n| \left| 1 - \frac{y^{2n}}{3!} + \frac{y^{4n}}{5!} - \ldots \right|
\end{aligned}
$$

and

$$
\begin{aligned}
|\sin(x^n)| &= \left| x^n - \frac{x^{3n}}{3!} + \frac{x^{5n}}{5!} - \ldots \right| \\
&= |x^n| \left| 1 - \frac{x^{2n}}{3!} + \frac{x^{4n}}{5!} - \ldots \right|
\end{aligned}
$$

$x, y, z \in \mathbb{C}$. We take $T(z)$ as $T_c(z)$, $x_0 = x$, $y_0 = y$ and $z_0 = z$ and assume that

(i) $\left| 1 - \frac{z^{2n}}{3!} + \frac{z^{4n}}{5!} - \ldots \right| \geq |\omega_1|$,

(ii) $\left| 1 - \frac{y^{2n}}{3!} + \frac{y^{4n}}{5!} - \ldots \right| \geq |\omega_2|$,

(iii) $\left| 1 - \frac{x^{2n}}{3!} + \frac{z^{4n}}{5!} - \ldots \right| \geq |\omega_3|$

where $|\omega_1|, |\omega_2|, |\omega_3| \in (0, 1]$ except the values of $x, y, z$ for which $|\omega_1| = |\omega_2| = |\omega_3| = 0$.

## 3. Escape Criteria for Complex Sine Functions

By utilizing four distinct celebrated iterations, we prove here an escape criterion for the transcendental complex sine function of the type $T(z) = \sin(z^n) + az + c$, $a, c \in \mathbb{C}$, $n \geq 2$.

**Theorem 1.** *Let $T_c(z) = \sin(z^n) + az + c$, $a, c \in \mathbb{C}$, $n \geq 2$, $|z| \geq |c| > \left( \frac{2(1+|a|)}{|\omega_1|} \right)^{\frac{1}{n-1}}$. If the sequence $\{z_k\}_{k \in \mathbb{N}}$ is Picard-iteration, then $|z_k| \to \infty$ as $k \to \infty$.*

**Proof.** Let $|z_{k+1}| = |T(z_k)|$. If $k = 0$,

$$
\begin{aligned}
|z_1| &= |T(z)| \\
&= |\sin(z^n) + az + c| \\
&\geq |\sin(z^n)| - |az| - |c| \\
&\geq |\sin(z^n)| - |a||z| - |z|, \ |z| \geq |c|, \\
&\geq |\omega_1||z^n| - |z|(1 + |a|) \\
&= |z|(1 + |a|)\left(\frac{|\omega_1||z^{n-1}|}{1 + |a|} - 1\right).
\end{aligned}
$$

Because $|\sin(z^n)| = \left|z^n - \frac{z^{3n}}{3!} + \frac{z^{5n}}{5!} - \dots\right| \geq |\omega_1||z^n|$, where $z \in \mathbb{C}$ except the values of $z$ for which $|\omega_1| = 0$, $|\omega_1| \in (0, 1]$. Thus, we get

$$
|z_1| \geq \frac{|z_1|}{1 + |a|} = |z|\left(\frac{|\omega_1||z^{n-1}|}{1 + |a|} - 1\right).
$$

For $k = 1$, we have

$$
\begin{aligned}
|z_2| &\geq |z_1|\left(\frac{|\omega_1||z_1^{n-1}|}{1 + |a|} - 1\right) \\
&\geq |z|\left(\frac{|\omega_1||z^{n-1}|}{1 + |a|} - 1\right)^2
\end{aligned}
$$

Because $|z_1| \geq |z|\left(\frac{|\omega_1||z^{n-1}|}{1+|a|} - 1\right)$ and $|z_1| \geq |z| \geq |c| > \left(\frac{2(1+|a|)}{|\omega_1|}\right)^{\frac{1}{n-1}}$, this implies that

$$
|z_1|\left(\frac{|\omega_1||z_1^{n-1}|}{1 + |a|} - 1\right) \geq |z|\left(\frac{|\omega_1||z^{n-1}|}{1 + |a|} - 1\right).
$$

On iterating till $k$th term,

$$
\begin{aligned}
|z_3| &\geq |z|\left(\frac{|\omega_1||z^{n-1}|}{1 + |a|} - 1\right)^3 \\
|z_4| &\geq |z|\left(\frac{|\omega_1||z^{n-1}|}{1 + |a|} - 1\right)^4 \\
&\quad . \\
&\quad . \\
&\quad . \\
|z_k| &\geq |z|\left(\frac{|\omega_1||z^{n-1}|}{1 + |a|} - 1\right)^k.
\end{aligned}
$$

Since $|z| \geq |c| > \left(\frac{2(1+|a|)}{|\omega_1|}\right)^{\frac{1}{n-1}}$, where $|\omega_1| \in (0, 1]$, this yields $\frac{|\omega_1||z^{n-1}|}{(1+|a|)} - 1 > 1$. Therefore, the orbit of $z$ tends to infinity, that is, $|z_k| \to \infty$ when $k \to \infty$. $\square$

**Theorem 2.** *Let $T_c(z) = \sin(z^n) + az + c$, $a, c \in \mathbb{C}$, $n \geq 2$, $|z| \geq |c| > \left(\frac{2(1+|a|)}{\alpha|\omega_1|}\right)^{\frac{1}{n-1}}$. If the sequence $\{z_k\}_{k \in \mathbb{N}}$ is Mann-iteration, then $|z_k| \to \infty$ as $k \to \infty$.*

**Proof.** Let $T_c(z) = \sin(z^n) + az + c$, $z_0 = z$ and

$$
|z_{k+1}| = |(1 - \alpha)z_k + \alpha T(z_k)|.
$$

If $k = 0$,

$$
\begin{aligned}
|z_1| &= |(1-\alpha)z + \alpha T(z)| \\
&= |(1-\alpha)z + \alpha(\sin(z^n) + az + c)| \\
&\geq \alpha|\sin(z^n)| - \alpha|az| - \alpha|c| - |(1-\alpha)z| \\
&\geq \alpha|\sin(z^n)| - \alpha|a||z| - \alpha|z| - |z| + \alpha|z|, \quad |z| \geq |c|, \\
&\geq \alpha|\omega_1||z^n| - |z||a| - |z|, \quad \alpha \in (0,1], \\
&= \alpha|\omega_1||z^n| - |z|(1+|a|) \\
&= |z|(1+|a|)\Big(\frac{\alpha|\omega_1||z^{n-1}|}{1+|a|} - 1\Big).
\end{aligned}
$$

Because $|\sin(z^n)| = \left|z^n - \frac{z^{3n}}{3!} + \frac{z^{5n}}{5!} - \dots\right| \geq |\omega_1||z^n|$, where $z \in \mathbb{C}$ except the values of $z$ for which $|\omega_1| = 0$, $|\omega_1| \in (0,1]$. Thus, we get

$$
|z_1| \geq \frac{|z_1|}{1+|a|} = |z|\Big(\frac{\alpha|\omega_1||z^{n-1}|}{1+|a|} - 1\Big).
$$

If $k = 1$, we have

$$
\begin{aligned}
|z_2| &\geq |z_1|\Big(\frac{\alpha|\omega_1||z_1^{n-1}|}{1+|a|} - 1\Big) \\
&\geq |z|\Big(\frac{\alpha|\omega_1||z^{n-1}|}{1+|a|} - 1\Big)^2.
\end{aligned}
$$

Since $|z_1| \geq |z|\Big(\frac{\alpha|\omega_1||z^{n-1}|}{1+|a|} - 1\Big)$ and $|z_1| \geq |z| \geq |c| > \Big(\frac{2(1+|a|)}{\alpha|\omega_1|}\Big)^{\frac{1}{n-1}}$, this implies that

$$
|z_1|\Big(\frac{\alpha|\omega_1||z_1^{n-1}|}{1+|a|} - 1\Big) \geq |z|\Big(\frac{\alpha|\omega_1||z^{n-1}|}{1+|a|} - 1\Big).
$$

On iterating until the $k$th term,

$$
\begin{aligned}
|z_3| &\geq |z|\Big(\frac{\alpha|\omega_1||z^{n-1}|}{1+|a|} - 1\Big)^3 \\
|z_4| &\geq |z|\Big(\frac{\alpha|\omega_1||z^{n-1}|}{1+|a|} - 1\Big)^4 \\
&\quad\quad\quad\quad \cdot \\
&\quad\quad\quad\quad \cdot \\
&\quad\quad\quad\quad \cdot \\
|z_k| &\geq |z|\Big(\frac{\alpha|\omega_1||z^{n-1}|}{1+|a|} - 1\Big)^k.
\end{aligned}
$$

Since $|z| \geq |c| > \Big(\frac{2(1+|a|)}{\alpha|\omega_1|}\Big)^{\frac{1}{n-1}}$, where $|\omega_1| \in (0,1]$, this yields $\frac{\alpha|\omega_1||z^{n-1}|}{(1+|a|)} - 1 > 1$. Therefore, the orbit of $z$ tends to infinity, that is, $|z_k| \to \infty$ when $k \to \infty$. $\square$

**Corollary 1.** *Let* $|z_m| > \max\Big\{|c|, \Big(\frac{2(1+|a|)}{\alpha|\omega_1|}\Big)^{\frac{1}{n-1}}\Big\}$, $m \geq 0$. *Since* $\frac{\alpha|\omega_1||z^{n-1}|}{(1+|a|)} - 1 > 1$, $|z_{m+k}| > |z|\Big(\frac{\alpha|\omega_1||z^{n-1}|}{(1+|a|)}\Big)^{m+k}$, *then* $|z_k| \to \infty$ *when* $k \to \infty$.

**Theorem 3.** *Let* $T_c(z) = \sin(z^n) + az + c$, $a, c \in \mathbb{C}$, $n \geq 2$, $|z| \geq |c| > \left(\frac{2(1+|a|)}{\alpha|\omega_1|}\right)^{\frac{1}{n-1}}$ *and*

$|z| \geq |c| > \left(\frac{2(1+|a|)}{\beta|\omega_2|}\right)^{\frac{1}{n-1}}$. *If the sequence* $\{z_k\}_{k \in \mathbb{N}}$ *is Ishikawa-iteration, then* $|z_k| \to \infty$ *as* $k \to \infty$.

**Proof.** Let $T(z) = \sin(z^n) + az + c$, $z_0 = z$ and

$$|y_k| = |(1-\beta)z_k + \beta T(z_k)|.$$

If $k = 0$,

$$
\begin{aligned}
|y_0| &= |(1-\beta)z_0 + \beta T(z_0)| \\
&= |(1-\beta)z + \beta(\sin(z^n) + az + c)| \\
&\geq \beta|\sin(z^n)| - \beta|az| - \beta|c| - |(1-\beta)z| \\
&\geq \beta|\sin(z^n)| - \beta|a||z| - \beta|z| - |z| + \beta|z|, \quad |z| \geq |c|, \\
&\geq \beta|\omega_1||z^n| - |z||a| - |z|, \quad \beta \in (0, 1], \\
&= \beta|\omega_1||z^n| - |z|(1 + |a|) \\
&= |z|(1 + |a|)\left(\frac{\beta|\omega_1||z^{n-1}|}{1 + |a|} - 1\right).
\end{aligned}
$$

Since $|\sin(z^n)| = \left|z^n - \frac{z^{3n}}{3!} + \frac{z^{5n}}{5!} - \dots\right| \geq |\omega_1||z^n|$, where $z \in \mathbb{C}$ except the values of $z$ for which $|\omega_1| = 0$, $|\omega_1| \in (0, 1]$. Thus, we get

$$|y| \geq \frac{|y|}{1 + |a|} = |z|\left(\frac{\beta|\omega_1||z^{n-1}|}{1 + |a|} - 1\right).$$

Since $|z| > \left(\frac{2(1+|a|)}{\beta|\omega_1|}\right)^{\frac{1}{n-1}}$, this provides that $|y^n| > |z|^n \left(\frac{\beta|\omega_1||z^{n-1}|}{1+|a|} - 1\right)^n \geq \beta|\omega_1||z|^n$. Now, for the next step of Ishikawa-iteration, we have

$$|z_{k+1}| = |(1-\alpha)z_k + \alpha T(y_k)|.$$

Again if $k = 0$,

$$
\begin{aligned}
|z_1| &= |(1-\alpha)z + \alpha T(y)| \\
&= |(1-\alpha)z + \alpha(\sin(y^n) + ay + c)| \\
&\geq \alpha|\sin(y^n)| - \alpha|ay| - \alpha|c| - |(1-\alpha)z| \\
&\geq \alpha|\sin(y^n)| - \alpha|a||y| - \alpha|z| - |z| + \alpha|z|, \quad |z| \geq |c|, \\
&\geq \alpha|\omega_2||y^n| - \alpha|y||a| - |z| \\
&\geq \alpha\beta|\omega_1||\omega_2||z^n| - \alpha\beta|\omega_1||z||a| - |z| \\
&= \alpha\beta|\omega_1||\omega_2||z^n| - |z||a| - |z|, \quad \alpha, \beta, \omega_1 \in (0, 1], \\
&= \alpha\beta|\omega_1||\omega_2||z^n| - |z|(1 + |a|) \\
&= |z|(1 + |a|)\left(\frac{\alpha\beta|\omega_1||\omega_2||z^{n-1}|}{1 + |a|} - 1\right).
\end{aligned}
$$

Since $|\sin(y^n)| = \left|y^n - \frac{y^{3n}}{3!} + \frac{y^{5n}}{5!} - \dots\right| \geq |\omega_2||y^n|$, where $z \in \mathbb{C}$ except the values of $z$ for which $|\omega_2| = 0$, $|\omega_2| \in (0, 1]$. In addition, $|y| \geq \beta|\omega_1||z|$. Thus, we get

$$|z_1| \geq \frac{|z_1|}{1 + |a|} = |z|\left(\frac{\alpha\beta|\omega_1||\omega_2||z^{n-1}|}{1 + |a|} - 1\right).$$

Now, on iterating up to $k$th term, we obtain

$$
\begin{aligned}
|z_2| &\geq |z|\left(\frac{\alpha\beta|\omega_1||\omega_2||z^{n-1}|}{1+|a|}-1\right)^2 \\
|z_3| &\geq |z|\left(\frac{\alpha\beta|\omega_1||\omega_2||z^{n-1}|}{1+|a|}-1\right)^3 \\
|z_4| &\geq |z|\left(\frac{\alpha\beta|\omega_1||\omega_2||z^{n-1}|}{1+|a|}-1\right)^4 \\
&\quad\cdot \\
&\quad\cdot \\
&\quad\cdot \\
|z_k| &\geq |z|\left(\frac{\alpha\beta|\omega_1||\omega_2||z^{n-1}|}{1+|a|}-1\right)^k.
\end{aligned}
$$

Since $|z| \geq |c| > \left(\frac{2(1+|a|)}{\alpha|\omega_1|}\right)^{\frac{1}{n-1}}$ and $|z| \geq |c| > \left(\frac{2(1+|a|)}{\beta|\omega_2|}\right)^{\frac{1}{n-1}}$, $|\omega_1|, |\omega_2| \in (0,1]$, this implies that

$$
|z| \geq |c| > \left(\frac{2(1+|a|)}{\alpha\beta|\omega_1||\omega_2|}\right)^{\frac{1}{n-1}}.
$$

Hence, $\frac{\alpha\beta|\omega_1||\omega_2||z^{n-1}|}{(1+|a|)}-1 > 1$ and the orbit of $z$ tends to infinity, that is, $|z_k| \to \infty$ when $k \to \infty$. $\square$

**Corollary 2.** *Let* $|z_m| > \max\left\{|c|, \left(\frac{2(1+|a|)}{\alpha|\omega_1|}\right)^{\frac{1}{n-1}}, \left(\frac{2(1+|a|)}{\beta|\omega_2|}\right)^{\frac{1}{n-1}}\right\}, m \geq 0.$ *Since* $\frac{\alpha\beta|\omega_1||\omega_2||z^{n-1}|}{(1+|a|)} - 1 > 1,$ *therefore,* $|z_{m+k}| > |z|\left(\frac{\alpha\beta|\omega_1||\omega_2||z^{n-1}|}{(1+|a|)}\right)^{m+k}.$ *Hence,* $|z_k| \to \infty$ *when* $k \to \infty.$

**Theorem 4.** *Let* $T_c(z) = \sin(z^n) + az + c$, $a, c \in \mathbb{C}$, $n \geq 2$, $|z| \geq |c| > \left(\frac{2(1+|a|)}{\alpha|\omega_1|}\right)^{\frac{1}{n-1}}$, $|z| \geq |c| > \left(\frac{2(1+|a|)}{\beta|\omega_2|}\right)^{\frac{1}{n-1}}$ *and* $|z| \geq |c| > \left(\frac{2(1+|a|)}{\gamma|\omega_3|}\right)^{\frac{1}{n-1}}.$ *If the sequence* $\{z_k\}_{k\in\mathbb{N}}$ *is Noor-iteration, then* $|z_k| \to \infty$ *as* $k \to \infty.$

**Proof.** Let $T(z) = \sin(z^n) + az + c$, $z_0 = z$ and

$$
|x_k| = |(1-\gamma)z_k + \gamma T(z_k)|.
$$

If $k = 0$,

$$
\begin{aligned}
|x_0| &= |(1-\gamma)z_0 + \gamma T(z_0)| \\
&= |(1-\gamma)z + \gamma(\sin(z^n) + az + c)| \\
&\geq \gamma|\sin(z^n)| - \gamma|az| - \gamma|c| - |(1-\gamma)z| \\
&\geq \gamma|\sin(z^n)| - \gamma|a||z| - \gamma|z| - |z| + \gamma|z|, \ |z| \geq |c|, \\
&\geq \gamma|\omega_1||z^n| - |z||a| - |z|, \ \gamma \in (0,1], \\
&= \gamma|\omega_1||z^n| - |z|(1+|a|) \\
&= |z|(1+|a|)\left(\frac{\gamma|\omega_1||z^{n-1}|}{1+|a|}-1\right).
\end{aligned}
$$

Since $|\sin(z^n)| = \left|z^n - \frac{z^{3n}}{3!} + \frac{z^{5n}}{5!} - \ldots\right| \geq |\omega_1||z^n|$, where $z \in \mathbb{C}$, except the values of $z$ for which $|\omega_1| = 0$, $|\omega_1| \in (0,1]$. Thus, we get

$$
|x| \geq \frac{|x|}{1+|a|} = |z|\left(\frac{\gamma|\omega_1||z^{n-1}|}{1+|a|}-1\right).
$$

Since $|z| > \left(\frac{2(1+|a|)}{\gamma|\omega_1|}\right)^{\frac{1}{n-1}}$, this gives that $|x^n| > |z|^n \left(\frac{\gamma|\omega_1||z^{n-1}|}{1+|a|} - 1\right)^n \geq \gamma|\omega_1||z|^n$.

Now, for the next step of Noor-iteration, we have

$$|y_k| = |(1-\beta)z_k + \beta T(x_k)|.$$

If $k = 0$,

$$
\begin{aligned}
|y_0| &= |(1-\beta)z + \beta T(x)| \\
&= |(1-\beta)z + \beta(\sin(x^n) + ax + c)| \\
&\geq \beta|\sin(x^n)| - \beta|ax| - \beta|c| - |(1-\beta)z| \\
&\geq \beta|\sin(x^n)| - \beta|a||x| - \beta|z| - |z| + \beta|z|, \quad |z| \geq |c|, \\
&\geq \beta|\omega_3||x^n| - \beta|x||a| - |z| \\
&\geq \beta\gamma|\omega_1||\omega_3||z^n| - \beta\gamma|\omega_1||z||a| - |z| \\
&= \beta\gamma|\omega_1||\omega_3||z^n| - |z||a| - |z|, \quad \beta, \gamma, \omega_1 \in (0,1], \\
&= \beta\gamma|\omega_1||\omega_3||z^n| - |z|(1+|a|) \\
&= |z|(1+|a|)\left(\frac{\beta\gamma|\omega_1||\omega_3||z^{n-1}|}{1+|a|} - 1\right).
\end{aligned}
$$

Since $|\sin(x^n)| = |x^n - \frac{x^{3n}}{3!} + \frac{x^{5n}}{5!} - \ldots| \geq |\omega_3||x^n|$, where $z \in \mathbb{C}$ except the values of $z$ for which $|\omega_3| = 0$, $|\omega_3| \in (0,1]$. In addition, $|x| \geq \gamma|w_1||z|$. Thus, we get

$$|y| \geq \frac{|y|}{1+|a|} = |z|\left(\frac{\beta\gamma|\omega_1||\omega_3||z^{n-1}|}{1+|a|} - 1\right).$$

Since $|y| > \left(\frac{2(1+|a|)}{\beta|\omega_2|}\right)^{\frac{1}{n-1}}$, this provides that $|y^n| > |z|^n \left(\frac{\beta\gamma|\omega_1||\omega_3||z^{n-1}|}{1+|a|} - 1\right)^n \geq \beta\gamma|\omega_1||\omega_3||z|^n$.

Now, for the next step of Noor-iteration, we have

$$|z_{k+1}| = |(1-\alpha)z_k + \alpha T(y_k)|.$$

Again if $k = 0$,

$$
\begin{aligned}
|z_1| &= |(1-\alpha)z + \alpha T(y)| \\
&= |(1-\alpha)z + \alpha(\sin(y^n) + ay + c)| \\
&\geq \alpha|\sin(y^n)| - \alpha|ay| - \alpha|c| - |(1-\alpha)z| \\
&\geq \alpha|\sin(y^n)| - \alpha|a||y| - \alpha|z| - |z| + \alpha|z|, \quad |z| \geq |c|, \\
&\geq \alpha|\omega_2||y^n| - \alpha|y||a| - |z| \\
&\geq \alpha\beta\gamma|\omega_1||\omega_2||\omega_3||z^n| - \alpha\beta\gamma|\omega_1||\omega_3||z||a| - |z| \\
&= \alpha\beta\gamma|\omega_1||\omega_2||\omega_3||z^n| - |z||a| - |z|, \quad \alpha, \beta, \gamma, \omega_1, \omega_3 \in (0,1], \\
&= \alpha\beta\gamma|\omega_1||\omega_2||\omega_3||z^n| - |z|(1+|a|) \\
&= |z|(1+|a|)\left(\frac{\alpha\beta\gamma|\omega_1||\omega_2||\omega_3||z^{n-1}|}{1+|a|} - 1\right).
\end{aligned}
$$

Since $|\sin(y^n)| = |y^n - \frac{y^{3n}}{3!} + \frac{y^{5n}}{5!} - \ldots| \geq |\omega_2||y^n|$, where $z \in \mathbb{C}$ except the values of $z$ for which $|\omega_2| = 0$, $|\omega_2| \in (0,1]$. In addition, $|y| \geq \beta\gamma|\omega_1||\omega_3||z|$. Thus, we get

$$|z_1| \geq \frac{|z_1|}{1+|a|} = |z|\left(\frac{\alpha\beta\gamma|\omega_1||\omega_2||\omega_3||z^{n-1}|}{1+|a|} - 1\right).$$

Now, on iterating up to $k$th term, we obtain

$$|z_2| \geq |z|\left(\frac{\alpha\beta\gamma|\omega_1||\omega_2||\omega_3||z^{n-1}|}{1+|a|} - 1\right)^2$$

$$|z_3| \geq |z|\left(\frac{\alpha\beta\gamma|\omega_1||\omega_2||\omega_3||z^{n-1}|}{1+|a|} - 1\right)^3$$

$$|z_4| \geq |z|\left(\frac{\alpha\beta\gamma|\omega_1||\omega_2||\omega_3||z^{n-1}|}{1+|a|} - 1\right)^4$$

$$.$$
$$.$$
$$.$$

$$|z_k| \geq |z|\left(\frac{\alpha\beta\gamma|\omega_1||\omega_2||\omega_3||z^{n-1}|}{1+|a|} - 1\right)^k.$$

Since $|z| \geq |c| > \left(\frac{2(1+|a|)}{\alpha|\omega_1|}\right)^{\frac{1}{n-1}}$, $|z| \geq |c| > \left(\frac{2(1+|a|)}{\beta|\omega_2|}\right)^{\frac{1}{n-1}}$ and $|z| \geq |c| > \left(\frac{2(1+|a|)}{\gamma|\omega_2|}\right)^{\frac{1}{n-1}}$, where $|\omega_1|, |\omega_2|, |\omega_3| \in (0,1]$ that is,

$$|z| \geq |c| > \left(\frac{2(1+|a|)}{\alpha\beta\gamma|\omega_1||\omega_2||\omega_3|}\right)^{\frac{1}{n-1}}.$$

Therefore,

$$\frac{\alpha\beta\gamma|\omega_1||\omega_2||\omega_3||z^{n-1}|}{(1+|a|)} - 1 > 1$$

and the orbit of $z$ tends to infinity that is, $|z_k| \to \infty$ when $k \to \infty$. $\quad\square$

**Corollary 3.** *Let* $|z_m| > \max\left\{|c|, \left(\frac{2(1+|a|)}{\alpha|\omega_1|}\right)^{\frac{1}{n-1}}, \left(\frac{2(1+|a|)}{\beta|\omega_2|}\right)^{\frac{1}{n-1}}, \left(\frac{2(1+|a|)}{\gamma|\omega_3|}\right)^{\frac{1}{n-1}}\right\}$, $m \geq 0$. *Since* $\frac{\alpha\beta\gamma|\omega_1||\omega_2||\omega_3||z^{n-1}|}{(1+|a|)} - 1 > 1$, *so* $|z_{m+k}| > |z|\left(\frac{\alpha\beta\gamma|\omega_1||\omega_2||\omega_3||z^{n-1}|}{(1+|a|)}\right)^{m+k}$. *Then,* $|z_k| \to \infty$ *as* $k \to \infty$.

**Remark 2.** (i) *The selection of the parameters in the above theorems is new and has not been studied till now in this perspective.*

(ii) *The Corollaries 1, 2 and 3 provide algorithms for exploring Julia sets of* $T_c$. *If* $|z| \leq |c|$, *we obtain the orbit of z. If* $|z_k|$ *lies in the exterior of the circle of radius for some k, then the orbit escapes meaning, thereby z is not inside the Julia sets. However, if* $|z_k|$ *does not exceed this bound, then utilizing the definitions of the Julia sets, we utilize these algorithms to generate fractals in the next section.*

## 4. Generation of Julia Sets

We attempt to obtain the non-classical Julia sets in four distinct orbits using Matlab 8.5.0 (R20015a). It has been observed from Figure 1 that the corresponding Julia sets for almost all the methods have no significant difference except in the Picard-iteration. However, as the iteration changes from one step to three steps, more beautiful results are noticed. We notice that, for Picard-iteration, we get a Julia set only in the neighborhood of 0. Moreover, the small change in any of the parameters gives the major change in the output. Therefore, we fix almost all the parameters and obtain the results for distinct iteration methods which are visible in the figures given below.

The parameters used in Figure 1 are as following in Table 1:

**Table 1.** The parameters used in Figure 1.

| | $a$ | $c$ | $\alpha$ | $\beta$ | $\gamma$ | $\omega_1$ | $\omega_2$ | $\omega_3$ | $n$ |
|---|---|---|---|---|---|---|---|---|---|
| *(i)* | 1.19802032 | $c = -1.89i$ | 0.17835 | 0.1675056 | 0.14323409 | 0.11025 | 0.115025 | 0.115025 | 3 |
| *(ii)* | 1.19802032 | $c = -1.89i$ | 0.17835 | 0.1675056 | - | 0.11025 | 0.115025 | - | 3 |
| *(iii)* | 1.19802032 | $c = -1.89i$ | 0.17835 | - | - | 0.11025 | - | - | 3 |

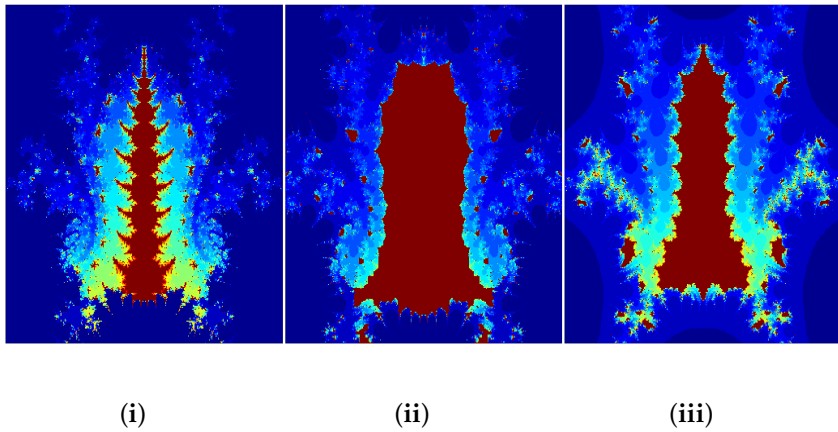

(**i**)　　　　　　　　(**ii**)　　　　　　　　(**iii**)

**Figure 1.** Cubic Julia sets in (**i**) Noor-orbit; (**ii**) Ishikawa-orbit; (**iii**) Mann-orbit.

The parameters used in Figure 2 are as following in Table 2:

**Table 2.** The parameters used in Figure 2.

|        | $a$        | $c$      | $\alpha$  | $\beta$   | $\gamma$     | $\omega_1$  | $\omega_2$   | $\omega_3$   |
|--------|------------|----------|-----------|-----------|--------------|-------------|--------------|--------------|
| (*i*)  | 1.19802032 | $-1.89i$ | 0.0077835 | 0.1675056 | 0.0814323409 | 0.07101025  | 0.079115025  | 0.078115025  |
| (*ii*) | 1.19802032 | $-1.89i$ | 0.0077835 | 0.1675056 | -            | 0.07101025  | 0.079115025  | -            |
| (*iii*)| 1.19802032 | $-1.89i$ | 0.0077835 | -         | -            | 0.07101025  | -            | -            |

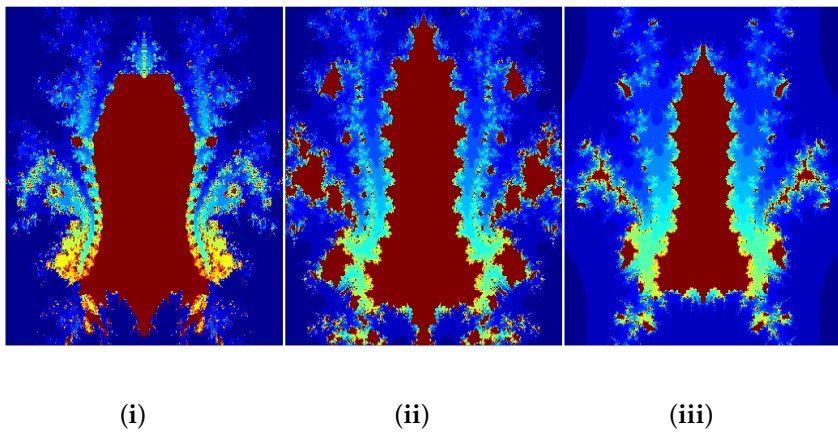

(**i**)　　　　　　　　(**ii**)　　　　　　　　(**iii**)

**Figure 2.** Cubic Julia sets in (**i**) Noor-orbit; (**ii**) Ishikawa-orbit; (**iii**) Mann-orbit.

The parameters used in Figure 3 are as following in Table 3:

**Table 3.** The parameters used in Figure 3.

|        | $a$          | $c$     | $\alpha$ | $\beta$    | $\gamma$     | $\omega_1$     | $\omega_2$ | $\omega_3$ |
|--------|--------------|---------|----------|------------|--------------|----------------|------------|------------|
| (*i*)  | 1.0229802032 | $-1.89$ | 0.077835 | 0.05675056 | 0.0814323409 | 0.000017101025 | 0.79115025 | 0.8115025  |
| (*ii*) | 1.0229802032 | $-1.89$ | 0.077835 | 0.05675056 | -            | 0.000017101025 | 0.79115025 | -          |
| (*iii*)| 1.0229802032 | $-1.89$ | 0.077835 | -          | -            | 0.000017101025 | -          | -          |

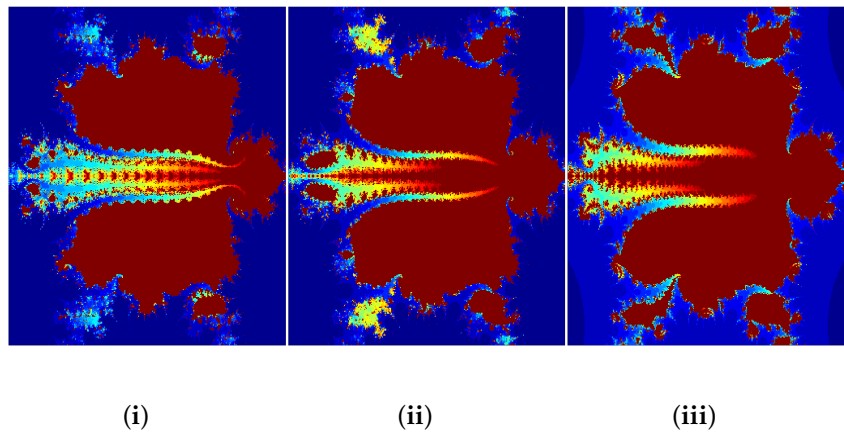

(**i**)         (**ii**)         (**iii**)

**Figure 3.** Cubic Julia sets in (**i**) Noor-orbit; (**ii**) Ishikawa-orbit; (**iii**) Mann-orbit.

The parameters with quintic ($n = 5$) used in Figure 4 are as following in Table 4:

**Table 4.** The parameters used in Figure 4.

|  | $a$ | $c$ | $\alpha$ | $\beta$ | $\gamma$ | $\omega_1$ | $\omega_2$ | $\omega_3$ |
|---|---|---|---|---|---|---|---|---|
| (*i*) | 1.0229802032 | −1.0089i | 0.077835 | 0.05675056 | 0.0814323409 | 0.000017101025 | 0.79115025 | 0.8115025 |
| (*ii*) | 1.0229802032 | −1.0089i | 0.077835 | 0.05675056 | - | 0.000017101025 | 0.79115025 | - |
| (*iii*) | 1.0229802032 | −1.0089i | 0.077835 | - | - | 0.000017101025 | - | - |

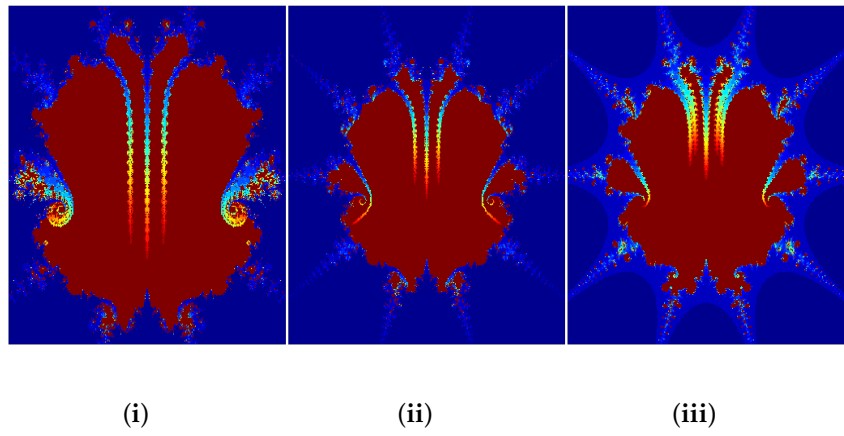

(**i**)         (**ii**)         (**iii**)

**Figure 4.** Quintic Julia sets in (**i**) Noor-orbit; (**ii**) Ishikawa-orbit; (**iii**) Mann-orbit.

The parameters used in Figures 5–7 with septic ($n = 7$) are as following in Table 5:

**Table 5.** The parameters used in Figures 5–7.

|  | $a$ | $c$ | $\alpha$ | $\beta$ | $\gamma$ | $\omega_1$ | $\omega_2$ | $\omega_3$ |
|---|---|---|---|---|---|---|---|---|
| (*i*) | 1.0229802032 | −1.89i | 0.077835 | 0.05675056 | 0.0814323409 | 0.000017101025 | 0.79115025 | 0.8115025 |
| (*ii*) | 1.0229802032 | −1.89i | 0.077835 | 0.05675056 | - | 0.000017101025 | 0.79115025 | - |
| (*iii*) | 1.0229802032 | −1.89i | 0.077835 | - | - | 0.000017101025 | - | - |

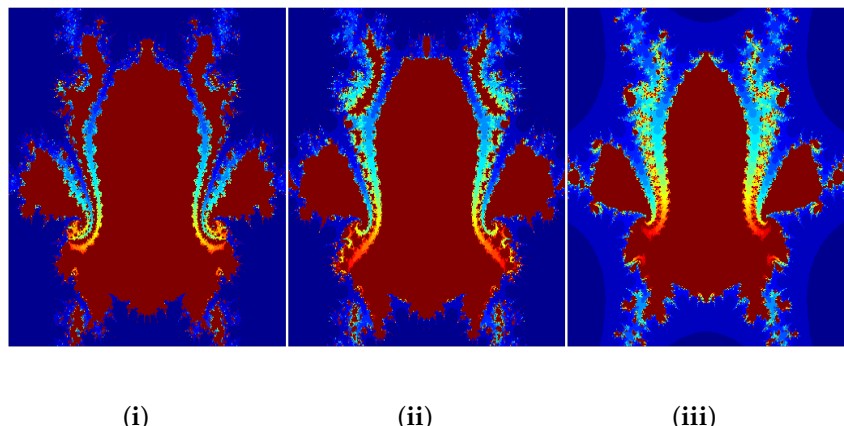

(**i**)                              (**ii**)                              (**iii**)

**Figure 5.** Cubic Julia sets in (**i**) Noor-orbit; (**ii**) Ishikawa-orbit; (**iii**) Mann-orbit.

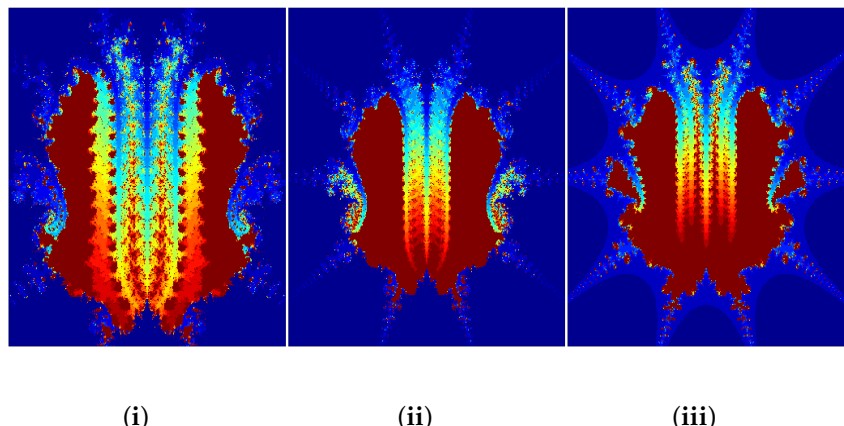

(**i**)                              (**ii**)                              (**iii**)

**Figure 6.** Quintic Julia sets in (**i**) Noor-orbit; (**ii**) Ishikawa-orbit; (**iii**) Mann-orbit.

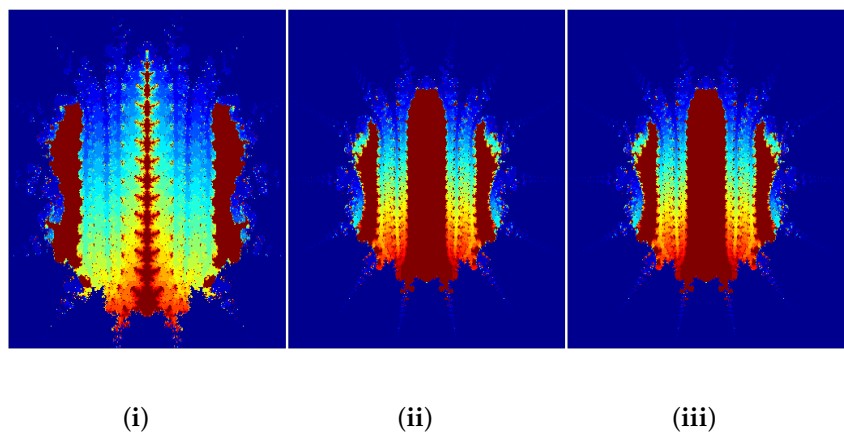

(**i**)                              (**ii**)                              (**iii**)

**Figure 7.** Septic Julia sets in (**i**) Noor-orbit; (**ii**) Ishikawa-orbit; (**iii**) Mann-orbit.

In Figures 8(i), (ii) fractals are looking like ants and in Figures 8(iii), 9–11, fractals are looking like Rangoli. As *n* increases the fractal takes a circular shape (see Figure 11 of the higher-order (*n* = 15) Julia set which looks like a colorful teething ring or circular saw or may be compared to glass paintings.

The parameters used in Figures 8–11 are as following in Table 6:

**Table 6.** The parameters used in Figures 8–11.

|  | $a$ | $c$ | $\alpha$ | $\beta$ | $\gamma$ | $\omega_1$ | $\omega_2$ | $\omega_3$ |
|---|---|---|---|---|---|---|---|---|
| (*i*) | 6.5 | 0 | 0.0097577835 | 0.11295675056 | 0.00975814323409 | 0.17101025 | 0.115025 | 0.8115025 |
| (*ii*) | 6.5 | 0 | 0.0097577835 | 0.11295675056 | - | 0.17101025 | 0.115025 | - |
| (*iii*) | 6.5 | 0 | 0.0097577835 | - | - | 0.17101025 | - | - |

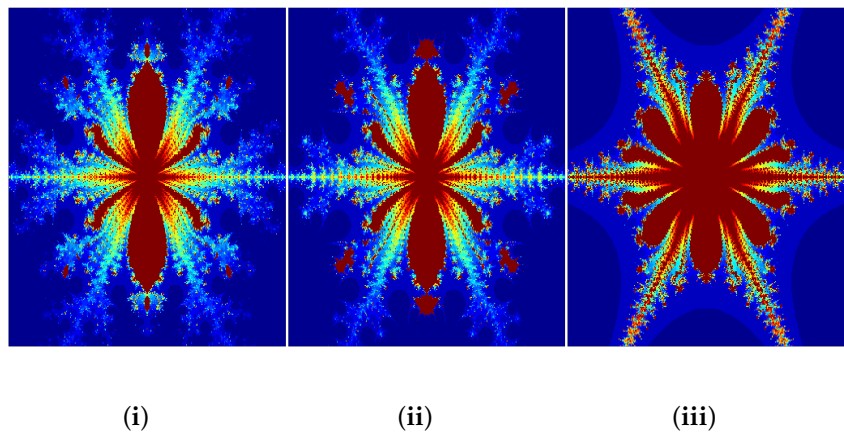

(**i**)         (**ii**)         (**iii**)

**Figure 8.** Cubic Julia sets in (**i**) Noor-orbit; (**ii**) Ishikawa-orbit; (**iii**) Mann-orbit.

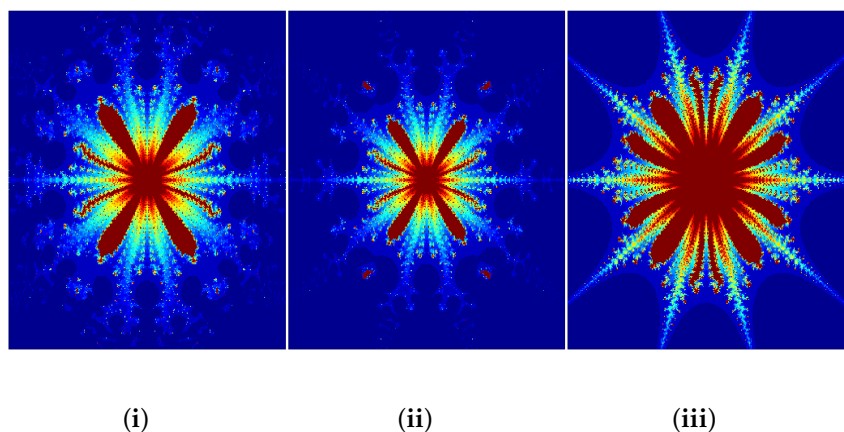

(**i**)         (**ii**)         (**iii**)

**Figure 9.** Quintic Julia sets in (**i**) Noor-orbit; (**ii**) Ishikawa-orbit; (**iii**) Mann-orbit.

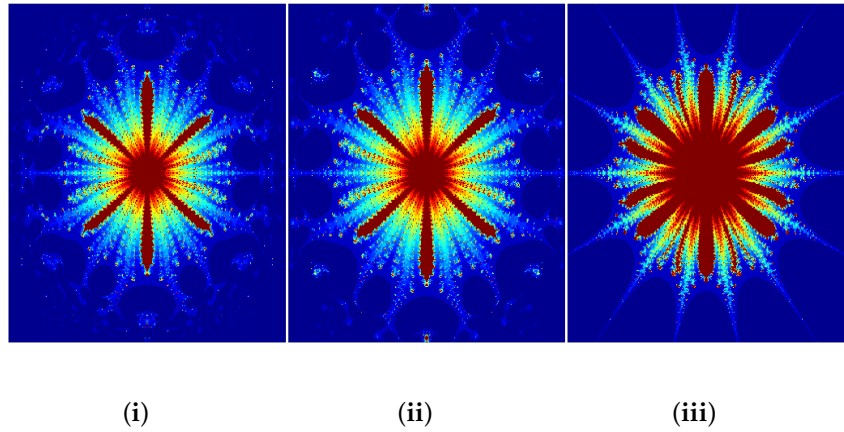

(**i**)         (**ii**)         (**iii**)

**Figure 10.** Septic Julia sets in (**i**) Noor-orbit; (**ii**) Ishikawa-orbit; (**iii**) Mann-orbit.

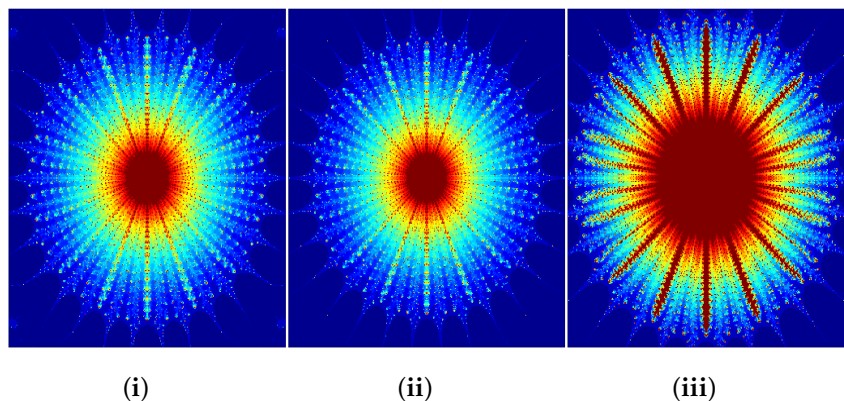

**(i)**          **(ii)**          **(iii)**

**Figure 11.** Higher-order Julia sets in (**i**) Noor-orbit; (**ii**) Ishikawa-orbit; (**iii**) Mann-orbit.

With the above parameters, we did not get Julia sets in Picard-orbit. Now, we explore some Julia sets in Picard-orbit also by suitably changing the parameters. In Figure 12(i)–(iii), fractals are looking like an earthen lamp (diya) with its own reflection along with real axis. In Figures 13–14, fractals are looking like Hibiscus and, in Figure 15, fractals are looking like Catharanthus Roseus (Rose Periwinkle).

The parameters used in Figures 12–15 are as following in Table 7:

**Table 7.** The parameters used in Figures 12–15.

|       | $a$  | $c$          | $\alpha$    | $\beta$      | $\gamma$    | $\omega_1$  | $\omega_2$     | $\omega_3$       |
|-------|------|--------------|-------------|--------------|-------------|-------------|----------------|------------------|
| (*i*)   | 1.65 | $-0.192975i$ | 0.12397835  | 0.125675056  | 0.04323409  | 0.17101025  | 0.00179115025  | 0.009118115025   |
| (*ii*)  | 1.65 | $-0.192975i$ | 0.12397835  | 0.125675056  | -           | 0.17101025  | 0.00179115025  | -                |
| (*iii*) | 1.65 | $-0.192975i$ | 0.12397835  | -            | -           | 0.17101025  | -              | -                |
| (*iv*)  | 1.65 | $-0.192975i$ | -           | -            | -           | 0.17101025  | -              | -                |

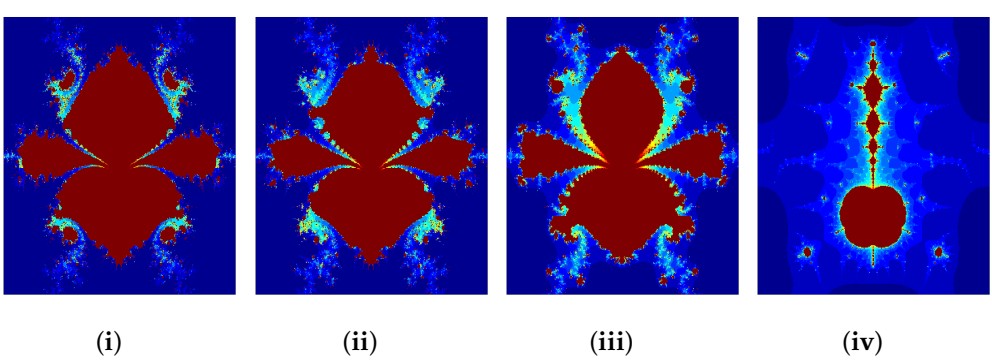

**(i)**       **(ii)**       **(iii)**       **(iv)**

**Figure 12.** Cubic Julia sets in (**i**) Noor-orbit; (**ii**) Ishikawa-orbit; (**iii**) Mann-orbit; (**iv**) Picard-orbit.

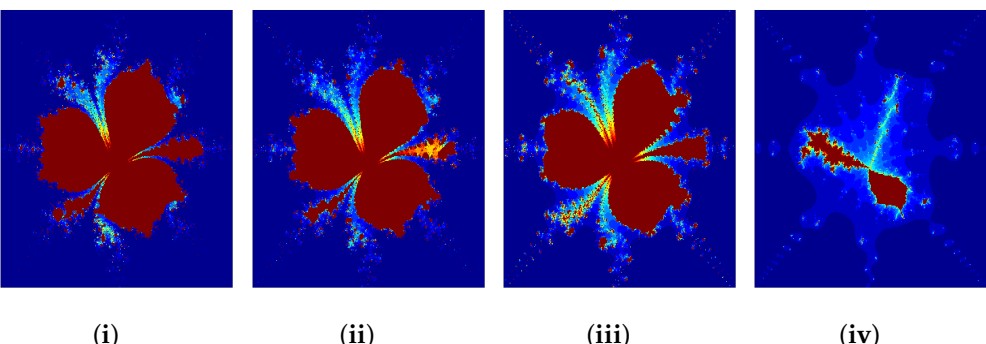

**(i)**       **(ii)**       **(iii)**       **(iv)**

**Figure 13.** Quintic Julia sets in (**i**) Noor-orbit; (**ii**) Ishikawa-orbit; (**iii**) Mann-orbit; (**iv**) Picard-orbit.

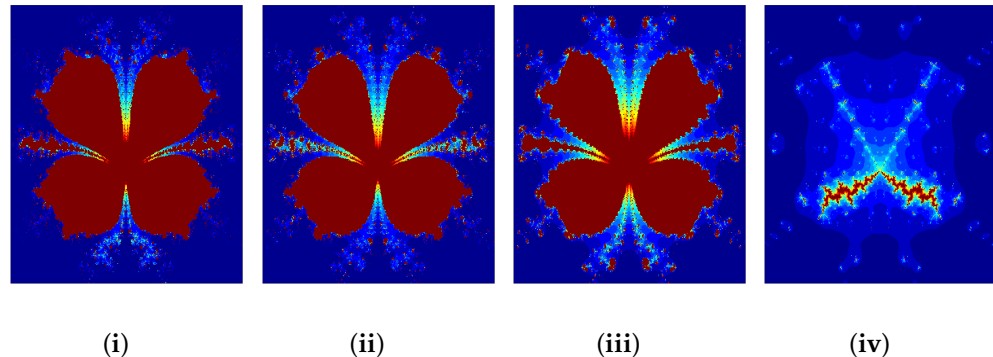

| (i) | (ii) | (iii) | (iv) |
|---|---|---|---|

**Figure 14.** Quintic Julia sets in (**i**) Noor-orbit; (**ii**) Ishikawa-orbit; (**iii**) Mann-orbit; (**iv**) Picard-orbit.

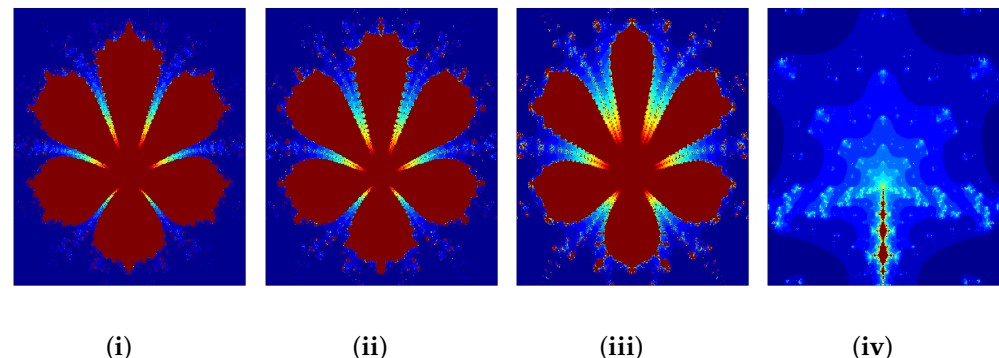

| (i) | (ii) | (iii) | (iv) |
|---|---|---|---|

**Figure 15.** Septic Julia sets in (**i**) Noor-orbit; (**ii**) Ishikawa-orbit; (**iii**) Mann-orbit; (**iv**) Picard-orbit.

The change in sign in parameters $a$ and $c$ leads to reflexive and rotational symmetry. However, the parameters $\alpha$, $\beta$, and $\gamma$ give beautiful texture when $\alpha < \beta < \gamma$. The vibrant color is highly dependent on the choice of $\alpha$, $\beta$, and $\gamma$. When $\alpha \geq \beta \geq \gamma$, it is observed that the fractals are connected. The parameters $\omega_1$, $\omega_2$, and $\omega_3$ are also used in getting the aura around each fractal. As $\omega_1$, $\omega_2$, and $\omega_3$ tend to zero, more beauty is added to the symmetrical pattern. It is seen that

1. the parameters $\alpha$, $\beta$, and $\gamma$ play a very important role in giving shape, size, and color to the fractals.
2. the convergence criteria derived for the fractals are also playing a very crucial role in giving resolution and richness of pixel in the fractals.
3. all the fractals developed in this paper are very novel, aesthetic, and pleasing as the function $f(z)$ contains the special kind of sine function in it.
4. the function $f(z) = \sin(z^n) + az + c$ carries lots of characteristics in it. Various combinations of parameters lead to a variety of fractals—some of which may be used to create fractal art on glass (to give stunning effects).
5. for the chosen function, we get the effects of flowers, ants, Rangoli, and glass painting in the fractals developed in the paper.

We displayed here only the zoomed version of each fractal because $\sin(z)$ is unbounded, and the fractals occupy the infinite area to lie in. However, because of unboundedness only on a real and imaginary axis, it can be visible.

## 5. Conclusions

In this work, we have utilized four different fixed point iterative methods to find convergence criteria for a special kind of transcendental complex function. It is worth mentioning here that transcendental functions, especially sine function, are often utilized to determine the solution of Laplace's equation, differential equations, and appears in the functional equation for the Gamma function, Riemann zeta-function, and so on.

Consequently, these are useful in distinct branches of science and engineering. We have generated variants of Julia sets and noticed that the size of fractals explored depends on the parameters $\alpha$, $\beta$, and $\gamma$, whereas the shape and symmetry depend on the parameter $a$, $b$ and $c$. In addition, as $n$ increases, the area occupied by the fractals decreases. It is fascinating to notice that some of our fractals represent the traditional Rangoli found in different parts of India and glass painting/art which is useful in interior decoration. Some represent ants and beautiful flowers (Hibiscus and Catharanthus Roseus) found in nature.

**Author Contributions:** Conceptualization, S.A., A.T., D.J.P. and M.S.; methodology, S.A. and A.T.; software, S.A. and D.J.P.; formal analysis, S.A. and A.T.; writing—original draft preparation, S.A., A.T. and D.J.P.; writing—review and editing, A.T. and M.S.; visualization, S.A. and A.T.; supervision, A.T. and D.J.P. All authors have read and agreed to the published version of the manuscript.

**Funding:** This research received no external funding.

**Institutional Review Board Statement:** Not applicable.

**Informed Consent Statement:** Not applicable.

**Data Availability Statement:** Not applicable.

**Acknowledgments:** The authors are grateful to the referees for their careful reading of the manuscript and for giving valuable remarks to improve it.

**Conflicts of Interest:** The authors declare no conflict of interest.

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
