# Peer review of "Fractals as Julia Sets of Complex Sine Function via Fixed Point Iterations"

_fractalfract, doi:10.3390/fractalfract5040272_

Round 1
Reviewer 1 Report
Report on the paper
entitled
\textbf{\textquotedblleft Fractals as Julia Sets of Complex Sine Function
via Fixed Point Iterations\textquotedblright }
submitted to
\textbf{\textquotedblleft Fractal and Fractional\textquotedblright }
\end{center}
\bigskip
In the paper under review the authors study on new variants of the Julia set
by developing the escape criteria for a transcendental complex function of
the type
\begin{equation*}
f(z)=\sin \left( z^{n}\right) +az+c,a,c\in
%TCIMACRO{\U{2102} }%
%BeginExpansion
\mathbb{C}
%EndExpansion
,\left( n\geq 2\right)
\end{equation*}%
where $z$ is a complex variable. For this purpose, four distinct fixed point
iterations are used. New algorithms are provided for exploring Julia sets
via four different fixed point iterations (the Picard iteration, the Mann
iteration, the Ishikawa-iteration, and the Noor-iteration). Also, the impact
of parameters on the deviation of dynamics, color, and appearance of
fractals are investigated by providing nice examples of fractals.
The results obtained in the paper are new and seem to be correct. The
structure of the paper is well-developed and easy to follow. Thus, I
recommend the \textbf{acceptance} of the paper in \textquotedblleft Fractal
and Fractional\textquotedblright .

Author Response
Thanks for acceptance paper in the present form. We have tried to improve "English language and style are fine/minor spell check required" as suggested.
Reviewer 2 Report
1. Line 85: “sin” may be better written as “sin”;
2. Line 175–176: There n is 5 in Figure 4?
3. Line 177–179: What is n in the so-called “Septic” case? What is the “Septic” meaning?
Author Response
The authors are thankful to reviewers for providing valuable comments and suggestions for the improvement of manuscript. The authors have tried to incorporate almost all possible comments and suggestions in the paper. If there are still some things left or not understood completely by authors (according to the point of view of reviewers), the authors are ready to get again comments and suggestions to improve this manuscript.
See below step-by-step responses of incorporating modifications on comments and suggestions.
- Line 85: “sin” may be better written as “sin”;
Response: Done
- Line 175–176: There n is 5 in Figure 4?
Response: n is 5 (Quintic). Done
- Line 177–179: What is n in the so-called “Septic” case? What is the “Septic” meaning?
Response: Septic means 7. Done
We have also tried to improve "English language and style are fine/minor spell check required" as suggested.
Reviewer 3 Report
Comments are enclosed in word document. Minor revision is needed.

Author Response
The authors are thankful to reviewers for providing valuable comments and suggestions for the improvement of manuscript. The authors have tried to incorporate almost all possible comments and suggestions in the paper. If there are still some things left or not understood completely by authors (according to the point of view of reviewers), the authors are ready to get again comments and suggestions to improve this manuscript. (see yellow color in manuscript)
See below step-by-step responses of incorporating modifications on comments and suggestions.
- Line 5. sin(z^n)+az+c where a, c ….
Response: Done.
- Line 6. Can fixed point iterations be replaced by fixed point iterative methods?
Response: Yes. Done.
- Line 16. Is there any reference for the sentence in line 16?
Response: Cited two references as [4,28]. However, it is an introductory well-known line.
- The introduction can be significantly improved by mentioning other applications of other iterative methods such as Euler’s method in forecasting for instance. Refer to the paper; Comparison of some forecasting methods for COVID-19. Alexandria Engineering Journal, 60,1, February 2021, pages 1565-1589. Add some literature from this paper and related papers to mention some real life applications of iterative methods.
Response: Done. Added references by writing some literature from these references [29,30,31].
- Line 28. Benoit B. Mandelbrot [4] drew …
Response: Done.
- Line 33. Iterates can be replaced by approximations.
Response: We have to check here behaviour under dynamics. So the authors feel that iterates is better word here.
- Page 3, line 72. Utilized by three Indian mathematicians …
Response: These are found in two Treatises written inside the bracket.
- Page 4, line 85. Write the equation on a single line and use normal font for sin.
Response: Done.
- When there is a formula or expression or equation or inequality, use a single line as far as possible such as: on page 4, lines 93-94; on page 5, lines 104-105.
Response: Done
10. Acknowledgement section can be included to mention if funding has been obtained for this research work.
Response: No funding is available for this research work. However, an acknowledgment section is not added in submission version of paper.
Introduction has also been improved.
The authors tried to remove the typos and stylistic errors noticed during the revision.